# Using a Directional Distance Function to Measure the Environmental Efficiency of International Liner Shipping Companies and Assess Regulatory Impact

Yi-Hui Liao [1] and Hsuan-Shih Lee [1,2,*]

1    Department of Shipping and Transportation Management, National Taiwan Ocean University, Keelung 202301, Taiwan
2    Department of Information Management, Ming Chuan University, Taipei 111005, Taiwan
*    Correspondence: hslee@email.ntou.edu.tw

**Abstract:** Maritime transport relies on a large amounts of fossil fuels. It provides cargo-carrying services but simultaneously emits enormous amounts of by-products such as $CO_2$, which cause climate change. The IMO has adopted mandatory measures to reduce the shipping industry's greenhouse gas emissions by at least 70% by 2050, relative to 2008. In this paper, we select 11 liner shipping companies as decision-making units (DMUs) that account for more than 80% of the world's shipping capacity. Utilizing the directional distance function, we estimate their environmental efficiency in 2019, 2020, and 2021. The directional vector serves to expand desired outputs and contract undesirable outputs. The larger the distance, the farther the evaluated unit is from the production frontier, and the less environmentally efficient it is assessed. This study compares the impact of environmental regulations on liner shipping based on two methods of dealing with undesirable outputs. Since the results indicate the low overall environmental efficiency of liner shipping, firms should strengthen their decarbonization efforts to achieve environmental efficiency goals. Moreover, the results also demonstrate that environmental regulations significantly impact liner shipping companies and that they need to reduce by-product outputs to comply with regulations.

**Keywords:** directional distance function; environmental efficiency; liner shipping; undesirable output

## 1. Introduction

Global warming has caused climate anomalies, and extreme climate phenomena have occurred frequently around the world. Carbon dioxide ($CO_2$) accumulated from human activities is the main cause of current global warming [1]. To curb climate change, world leaders at the UN Climate Change Conference (COP21) in Paris signed the Paris Agreement in 2015. Its goals are the reduction of global greenhouse gas (GHG) emissions and the limitation of global temperature rise in this century to 2 °C, or preferably 1.5 °C.

Shipping is responsible for global cargo transportation and trade facilitation. According to the fourth greenhouse gas study released by the International Maritime Organization (IMO) in 2020, the share of shipping emissions in global anthropogenic emissions in 2018 accounted for 2.89% of the total. At the 72nd meeting of the Marine Environment Protection Committee (MEPC) in 2018, the IMO adopted an initial strategy for reducing GHGs from international ships in response to the Paris Agreement. The initial strategy requires international shipping to reduce its carbon intensity by 40% by 2030 and toward 70% by 2050, relative to 2008 levels [2,3]. The IMO has adopted short-term, medium-term, and long-term measures according to these timelines. Short-term measures include technical and operational actions. Medium-term and long-term measures include market-based measures (MBMs) and alternative energy sources. The 76th meeting of the MEPC in 2021 approved draft amendments to the International Convention for the Prevention of Pollution from Ships (MARPOL Annex VI), which combine technical and operational approaches.

The amendments set a clear standard for ship carbon emissions and took effect on 1 January 2023. The Energy Efficiency Existing Ships Index (EEXI) requires shipping companies to evaluate whether the vessels in their fleets can meet the new regulations. If vessels cannot meet the requirements of EEXI, they should implement carbon reduction measures and install energy-saving devices. The new carbon intensity indicator (CII) aims to evaluate the operational energy efficiency of ships. It applies to ships of 5000 gross tonnage (GT) and above. Vessels will receive a rating for their energy efficiency annually. These ratings range from A to E, with A being the best [4].

Port assistance is also crucial in carbon reduction actions. The IMO encourages voluntary cooperation between ports and the shipping sector to meet carbon reduction targets (MEPC.323 (74)). Ports represent a mix of public and private sectors, and port authorities make regulations and standards to conform with MARPOL. Simultaneously, they provide incentives or grants, tariff changes, concessions, and lease contracts [5]. In the private sector, port operators adopt technical and operational approaches to mitigate GHG emissions associated with ship-port interface activities, such as providing onshore power supply (OPS), supplying alternative fuels, shortening ship turnaround time, providing hull cleaning, and applying hardware and software facilities. Additionally, incentive schemes are also among the management tools that promote the mitigation of GHG emissions from international ships. Incentives that encourage ships to adopt green shipping measures are divided into industry-initiated and international organization-initiated. For example, the Environmental Shipping Index (ESI) is a project initiated by the World Ports Climate Initiative and the International Association of Ports and Harbors (IAPH). The evaluation targets are sulfur oxides (SOx), nitrous oxides (NOx), $CO_2$ emissions from ocean-going vessels, and shore power usage. When qualifying vessels' berth at participating ports, the ports will grant discounts on port dues based on vessels' ESI scores [6,7].

The trend in carbon reduction policies is clear, but the shipping industry can achieve $CO_2$ mitigation through various methods. Romano et al. [8] summarized decarbonization measures in marine transportation including innovative technology, renewable energy sources, alternative fuels, and policy development. The perspective of liner shipping companies (LSCs) involves obtaining a certain number of ships and then focusing on transportation services, operating liner schedules, and rotating through fixed ports. LSCs also can manage vessel deployments, vessel schedules, and ship routing, and utilize speed optimization to achieve $CO_2$ reductions [9–11]. LSCs facing increasingly strict environmental regulations must demonstrate environmental efficiency and utilize green management techniques to help maintain environmental sustainability [12]. These studies have examined decarbonization and $CO_2$ mitigation approaches, various simulations to optimize energy efficiency or design voyage scheduling models, and speed optimization within marine transportation. However, few authors have discussed the environmental performance of LSCs in reducing $CO_2$ emissions, and previous studies have focused on shipping companies' financial and operational performance. Therefore, to fill the gap in the previous literature, this study's purpose is to evaluate the environmental performance of LSCs in implementing $CO_2$ reduction measures, and the directional distance function (DDF) is used to evaluate the environmental efficiency of LSCs from 2019 to 2021. Additionally, $CO_2$ is regarded as a normal output and different models are used to assess $CO_2$ emissions and identify the appropriate production boundaries and the most efficient companies. The results of this study can help LSCs identify strengths and weaknesses and help plan for future improvement.

The remainder of this paper is structured as follows: Section 2 presents literature reviews. Section 3 describes the materials and methodology. Section 4 presents the results. The last section provides conclusions and implications for future research.

## 2. Literature Review

Enterprises use resources and energy to produce products and provide services, which inevitably produce by-products and waste. While pursuing economic growth, enterprises

should also consider environmental sustainability and development. While sailing to and from fixed ports, ships emit GHGs and generate pollution, especially exhaust gases produced while idling, affecting the residents near ports. GHG emissions are related to the type of fuel used by ships [13], and the primary fuels used by the global fleet are marine diesel oil (MDO), heavy fuel oil (HFO), and liquefied natural gas (LNG), among which HFO and MDO dominate the market. HFO is inexpensive and energy-intensive, but also emits copious amounts of $CO_2$ [14].

A great deal of literature has presented approaches to reducing GHG emissions [8,15–17]. Bouman et al. [16] presented six primary measures and 22 individual approaches. Factors including ship size, biofuels use, capacity utilization, and speed optimization can increase the potential for $CO_2$ mitigation. Vessel size prioritizes achieving economies of scale based on transport, per unit of cargo. Woo et al. [18] simulated the effects of slow steaming in liner shipping and found that slow steaming and the enlargement of vessel size could reduce $CO_2$ emissions. Other studies have examined the use of alternative fuels such as LNG, biodiesel, methanol, hydrogen, ammonia, and ethanol. Currently, shipping companies face difficulties obtaining alternative fuels, except for LNG. These problems require technological innovations to overcome [19]. Presently, LNG, which reduces $CO_2$ emissions by 20–30%, is a great transitional choice to replace MDO and HFO [20]. Qi and Sog [11] investigated the vessel schedule, modeling port handling uncertainty. Golnar and Beskovnik [21] used a multi-criteria approach with the AHP-DEA method to evaluate a sustainable intermodal transport chain. Based on the weighting of criteria, users selected the prioritization of the most efficient transport combination. Relevant reduction measures focus not only on technological and operational measures but also on MBMs. Psaraftis [22] presented a detailed review of MBMs. Shi [23] proposed adopting MBMs to reduce GHG emissions from international shipping. The existing literature focuses on energy efficiency, a comprehensive look at measures, practices and policies, alternative fuels, and alternative resources. However, LSCs' environmental efficiency is discussed less frequently.

In previous studies, shipping companies' efficiency assessments were found to stress financial and operational efficiencies [24–31]. Panayides et al. [24] used stochastic frontier analysis (SFA) and data envelopment analysis (DEA) to examine the relative market and operating efficiencies of major international maritime firms. Lun and Marlow [25] considered that within liner shipping operations, capacity management was a crucial factor that affected performance. The study applied the traditional DEA with constant returns to scale model and used two inputs (shipping capacity and operating cost) and two outputs (profit and revenue) to assess the effect of scale operations on liner shipping in 2008. Bang et al. [26] combined DEA and Tobit regression to investigate the relative efficiencies of 14 global shipping line companies in 2008 in terms of operational and financial performance. Gutiérrez et al. [27] applied a bootstrap DEA to evaluate the effects on container shipping lines during the 2008 economic crisis. Chao [28] comprised a multi-stage DEA model and a fuzzy analytical hierarchical process (FAHP) to evaluate the operational efficiencies of 15 global LSCs in 2012. Chao et al. [29] proposed a dynamic network DEA with shared inputs to evaluate the efficiencies of 13 global container shipping companies from 2013 to 2015. Gong et al. [30] applied a slack-based measure (SBM) DEA model to evaluate the economic and cargo efficiencies of shipping companies both with and without the negative impact of emissions. Wang et al. [31] noticed the corporate social responsibility of cruise lines, using Super SBM and DEA-based Malmquist to measure the environmental efficiency of Carnival and its subsidiaries from 2010 to 2015. Except for Gong et al. [30] and Wang et al. [31], these referenced studies did not consider the undesirable output during production processes.

The implementation of $CO_2$ reduction by the maritime transport industry is an international concern and the IMO's imperative environmental policy. Therefore, when evaluating LSCs' performance, undesirable outputs should be considered. In the present study, $CO_2$ emissions from ships are treated as undesirable outputs and the DDF is used to measure LSCs' environmental efficiencies. The advantage of this method allows for

the possibility of increasing desirable output and simultaneously decreasing undesirable output. Shephard [32] first proposed the distance function to measure the technical efficiency of the distance along the direction of a ray passing through the DMU. The output model expands from the rated unit to the outside, and the input model shrinks toward the origin point. Chambers et al. [33] constructed a DDF based on Luenberger's (1992) shortage function that had been proposed by Shephard's input and output distance function. The maximum distance could advance in a direction $(-x, y)$, which was an appointed direction in the production possibility set. The direction vector was to increase the desirable output and decrease the undesirable output simultaneously, and the distance was the basis of measuring efficiency. Another concept was joint production. Färe et al. [34] suggested that the production process produced a desired product, but also produced an undesirable one. This by-product could not be free or discarded at no cost. Instead, coping with these by-products was expensive. Therefore, traditional productivity performance analyses which did not include undesirable outputs may result in efficiency misjudgments. Chung et al. [35] extended Färe's model [34] and applied a Malmquist-Luenberger productivity index model based on a DDF whose vector contracts undesired outputs and expands desired outputs. The index offered companies options for measuring productivity when facing environmental regulations. Färe et al. [36] established regulated and unregulated models using DDF and evaluated the pollution abatement costs of US power plants in 1994 and 1995. The difference in DDF value between regulated and unregulated is regarded as the loss of the desirable output due to decreased undesirable output, and is also treated as the cost of pollutant reduction.

Färe et al. [37] modeled an environmental direction distance output function in which the directional vector maintained inputs while expanding good outputs and reducing bad outputs. Watanabe et al. [38] applied traditional DEA and DDF to analyze the efficiency performance of China's industries from 1994 to 2002. The results showed that industrial efficiency in Chinese provinces could be overestimated without evaluating undesirable outputs. Mandal and Madheswaran [39] examined the Indian cement industry and applied two models. One treated $CO_2$ emissions as inputs and the other considered pollutants as undesirable outputs. The directional vector served to increase the desirable outputs and decrease the undesirable ones without increasing the inputs. Oggioni et al. [40] evaluated the eco-efficiency of the cement industry in 21 countries and compared pollutants as inputs or outputs. They measured the technology of the global cement industry in association with input reductions. The results indicated that the country's environmental efficiencies were dependent on mandatory environmental regulations and investment in alternative energy sources. Tovar and Wall [41] adopted the DDF method to estimate the environmental efficiencies of Spanish port authorities with variable returns to scale using two different vectors; one increased the desirable outputs while reducing undesirable ones, and the other reduced the undesirable outputs while maintaining the desirable ones.

In summary, the DDF method with the concept of joint production uses linear programming to identify the production boundary and evaluate the relative efficiency of the DMUs. Analyzing environmental efficiency performance can utilize different directional vectors, such as limiting input, reducing undesirable output, or increasing desired ones. For example, Zhang and Choi [42] integrated the DDF used in energy and environmental research and provided recommendations for model selection.

## 3. Materials and Methodology

### 3.1. Materials

This study aims to evaluate the environmental efficiency of the world's major LSCs from 2019 to 2021. According to the Alphaliner website [43], the world's major firms are A.P.Moller-Maersk (Maersk), Mediterranean Shipping Company (MSC), CMA CGM Group, COSCO Group, Ocean Network Express (ONE), Hapag-Llyod, Evergreen Line, Yang Ming Marine Transport Corp., Wan Hai Lines, HMM Co Ltd., and ZIM. These LSCs maintain fleets with huge shipping capacity and account for more than 80% of global capacity. DMUs

provide shipping services. The input variables are the resources they used, and the output variables are the resources that are converted into products or labor services. The objective of this paper is to evaluate the environmental efficiency of DMUs. The variables were selected according to the available data but also considered the environmental efficiency that DMUs identify.

This paper uses the DDF method to evaluate the environmental efficiency of LSCs. The firms' data were collected from corporate social reports (CSR), financial reports, and annual reports in 2019, 2020, and 2021. Efficiency analyses of shipping companies often treat the number of vessels and transportation capacity as input variables [25,26,28,30], however, we considered how the size of ships and $CO_2$ emissions vary greatly. Additionally, shipping capacity represents the LSCs' ability to supply; therefore, shipping capacity was selected as an input variable. Fuel consumption is an indispensable input cost in the production process of ships and is one of the primary undesirable output factors of production. MSC's reports do not disclose fuel consumption levels, but since it plays a pivotal role in global transportation capacity due to its vessels and transportation capacity, its GHG emissions cannot be ignored. Therefore, the calculation of MSC fuel oil usage was based on IMO resolution MEPC 245(66) "2014 Guidelines on the Method of Calculation of the Attained Energy Efficiency Design Index (EEDI) for New Ships". Thus, $CO_2$ Emissions = Fuel consumption $\times$ Emission factor, heavy oil emission factor (tCO2/tFuel) 3.114. Furthermore, the unit of fuel quantity used by HMM's CRS was described using Mwh. For the convenience of calculation, this article converts 1 Mwh to 0.08598 tons. Output variables were the volume of the containers carried and $CO_2$ emissions. Therefore, input variables in this paper included capacity and fuel usage. The volume of the containers carried represents the actual volume of the containers transported by each LSC, so it is desirable output. Undesirable output is $CO_2$, which was taken from SCOPE 1 in the CSR, which is the direct emission of gas generated by the combustion of ship fuels. While ship gas emissions include not only $CO_2$, but also NOx, SOx, and other pollutants, this paper chose $CO_2$ as the undesirable output because of the carbon reduction goal. The descriptive statistics of input and output are shown in Table 1.

**Table 1.** Descriptive statistics of the variables.

| | Statistics Analysis | Input Capacity (TEU) | Fuel (Scope 1) | Desirable Output Cargo Carried (TEU) | Undesirable Output $CO_2$ (tons) |
|---|---|---|---|---|---|
| 2019 | Maximum | 4,132,000 | 11,173,000 | 26,592,000 | 36,204,000 |
| | Minimum | 250,900 | 976,401 | 2,811,000 | 2,199,110 |
| | Range | 3,881,100 | 10,196,599 | 23,781,000 | 34,004,890 |
| | Mean | 1,725,679 | 4,616,600 | 12,405,769 | 14,179,979 |
| | Standard Deviation | 1,359,280 | 3,514,757 | 8,360,093 | 11,424,998 |
| 2020 | Maximum | 4,081,000 | 10,368,000 | 25,268,000 | 33,902,000 |
| | Minimum | 323,357 | 939,400 | 2,841,000 | 2,931,720 |
| | Range | 3,757,643 | 9,428,600 | 22,427,000 | 30,970,280 |
| | Mean | 1,799,595 | 4,406,623 | 12,209,357 | 13,680,750 |
| | Standard Deviation | 1,342,193 | 3,387,274 | 8,259,234 | 10,930,302 |
| 2021 | Maximum | 4,313,568 | 11,083,000 | 26,178,000 | 36,863,000 |
| | Minimum | 260,000 | 1,258,038 | 3,814,000 | 3,989,792 |
| | Range | 4,053,568 | 9,824,962 | 22,364,000 | 32,873,208 |
| | Mean | 1,914,391 | 4,780,623 | 12,570,318 | 14,920,446 |
| | Standard Deviation | 1,423,268 | 3,676,115 | 8,522,306 | 11,965,438 |

### 3.2. Methodology

LSCs provide maritime transportation services carrying goods, but also emit $CO_2$. According to Chung et al. [35] and Färe et al. [37], the DDF with joint production mode is applied to evaluate $CO_2$ output. First, we define the joint production technology denoting

inputs as $x \in R_+^N$, desirable outputs as $y \in R_+^M$, and undesirable outputs as $b \in R_+^j$. The technology set represents all technologically feasible relationships between inputs and outputs. The expressions are as follows:

$$T = \{(x, y, b)\ : x\ can\ produce\ (y, b)\} \tag{1}$$

$$(x, y, b) \in T \iff (y, b) \in P(x) \tag{2}$$

Let the output set $P(x)$ denote the set of desirable and undesirable outputs that are jointly produced from the input vector $x$.

$$P(x) = \{(y,\ b) : x\ can\ produce\ (y, b)\ \} \tag{3}$$

The output set must meet the following environmental properties. First, following Färe et al. [37], desirable and undesirable outputs are assumed to be null-jointness. This means that a firm that produces desirable outputs must also produce undesirable ones simultaneously. Thus, if the firm does not want to produce undesirable outputs, the only way to do so is to not manufacture desired outputs.

$$(y,\ b\ ) \in P(x)\ and\ b = 0,\ then\ y = 0 \tag{4}$$

The second assumption is that the desirable output is strongly disposable.

$$(y,\ b) \in P(x)\ and\ y' \leq y\ imply\ (y', b) \in P(x) \tag{5}$$

Third, desirable and undesirable outputs are considered jointly weakly disposable. This implies that it is feasible to reduce outputs proportionally by $\theta$ and means companies reduce a bad output with a fixed percentage reduction of a good output. Therefore, it is impossible to reduce undesirable output without reducing a desired one.

$$P(y,\ b) \in P(x)\ and\ 0 \leq \theta \leq 1\ then\ (\theta b, \theta y) \in P(x) \tag{6}$$

In contrast, it is assumed that firms face unregulated environmental technology. Outputs are reduced non-proportionally and undesirable outputs are freely disposable.

$$(y,\ b) \in P(x)\ and\ (y',\ b') \leq (y,\ b)\ imply\ (y',\ b') \in P(x) \tag{7}$$

Let $g = (g_y, -g_b)$ be the direction vector. The vectors represent expanding the desirable outputs in the $g_y$ direction and contracting the undesirable outputs in the $g_b$ direction. $\beta$ is the maximum feasible enlargement of desirable output and contraction of undesirable output in the direction of the vector of $(y + \beta g_y, b - \beta g_b)$. When the $\beta$ value is 0, the evaluated unit is on the leading edge of production, which is equivalent to $\theta$ value = 1, the standard DEA mode. This indicates that the DMU is efficient. In contrast, when the $\beta$ value is higher than zero, the farther the DMU is from the efficiency frontier, the more inefficient the environmental technology is.

$$\overrightarrow{D_T}(x, y, b; g_y, -g_b) = max\ \{\beta : (y + \beta g_y, b - \beta g_b\ ) \in T\} \tag{8}$$

Utilizing environmental technology, we assume that undesirable outputs are weakly disposable and that reduction in undesirable output is impossible without reducing desirable output. The model of DDF with constant returns to scale and weak disposability is as follows. To guarantee $\beta^{k'} \leq 1$, we let $g_y = y_{k'}$ and $g_b = b_{k'}$. $Z_k$ represent the weight of the $k^{th}$. Model (9) becomes the following:

$$\overrightarrow{D^w}(x, y, b\ ; y_{k'}, -b_{k'}) = max\beta^{k'}$$

$$s.t. \sum_{k=1}^{K} Z_k y_{km} \geq y_{k'm} + \beta^{k'} y_{k'm} \; , \quad m = 1, \dots M$$

$$\sum_{k=1}^{K} b_{kj} = b_{k'j} - \beta^{k'} b_{k'j}, \quad j = 1, \dots J \tag{9}$$

$$\sum_{k=1}^{K} Z_k x_{kn} \leq x_{k'n} \, , \quad n = 1, \dots, N,$$

$$Z_k \geq 0 \; k = 1, \dots, K$$

Assuming the undesirable output has strong disposability, the undesirable output can be freely disposed of without cost. Model (10) becomes the following:

$$\overrightarrow{D^w}(x, y, b \, ; y_{k'}, -b_{k'}) = \max \beta^{k'}$$

$$s.t. \sum_{k=1}^{K} Z_k y_{km} \geq y_{k'm} + \beta^{k'} y_{k'm} \; , \quad m = 1, \dots M$$

$$\sum_{k=1}^{K} b_{kj} \geq b_{k'j} - \beta^{k'} b_{k'j}, \quad j = 1, \dots J \tag{10}$$

$$\sum_{k=1}^{K} Z_k x_{kn} \leq x_{k'n} \, , \quad n = 1, \dots, N,$$

$$Z_k \geq 0 \; k = 1, \dots, K$$

The difference between model (9) and model (10) is the inequality of undesirable output. It implies the way of treating undesirable output. Undesirable output in weak disposability must comply with environmental protection regulations. And strong disposability assumed that undesirable output is not mandatory according to regulations. All inputs pursue the maximization of desirable outputs and the emission of by-products without occurring any costs. While assuming LSCs with strong disposability may be unrealistic, this article attempted to emphasize that LSCs used copious amounts of fuel to carry the maximum number of containers without actively pursuing $CO_2$ mitigation.

According to Picazo-Tadeo et al. [44], the values of the two hypotheses represent the impact of environmental regulation on LSCs. Expression (11) represents the pollution abatement cost (PAC). The difference indicates the distance that the evaluated unit $k'$ projects onto both the regulated and unregulated efficient frontiers. The value implies the opportunity cost of regulation. In other words, the loss of good output due to regulation.

$$\text{PAC} = \overrightarrow{D^s}(x, y, b \, ; y_{k'}, -b_{k'}) - \overrightarrow{D^w}(x, y, b \, ; y_{k'}, -b_{k'}) \tag{11}$$

Under two assumptions of environmental regulations, the value of $RI_{k'm}$ is a regulatory impact from environmental regulations. It implies that LSCs are dedicated to reducing $CO_2$ emissions and the loss is caused by their transition. If $RI_{k'm} = 0$, it signifies that the environmental regulations have no impact on the company. In contrast, if the value $> 0$, environmental regulations impact companies.

$$RI_m^{k'} = \left[ \overrightarrow{D^s}(x, y, b \, ; y_{k'}, -b_{k'}) - \overrightarrow{D^w}(x, y, b \, ; y_{k'}, -b_{k'}) \right] y_{k'm} \tag{12}$$

The proposed DDF method is briefly illustrated in Figure 1. The DDF methodological steps are as follows:

(1) Goal setting: The objective of this paper was to evaluate LSCs' environmental efficiency. DMUs that obtain high shipping capacity and have pivotal roles globally

were selected. For the collection of input and output variables, empirical data and environmental efficiency had to match.

(2)  Denote environmental properties and build the DDF model: denote "null-jointness". Desirable outputs were produced along with undesirable outputs. This implied that LSCs could not produce desirable outputs without producing undesirable outputs. Desirable outputs were freely disposable. Next, there were two assumptions about undesirable outputs. One, desirable and undesirable outputs were weakly disposable under a regulated environment. Thus, the disposal of undesirable outputs was not a free activity. The other was an unregulated environment. Undesirable outputs were strongly disposable. One could dispose of undesirable outputs without any cost.

(3)  Calculate the value of the DDF under the assumptions of the weak disposability and strong disposability of undesirable outputs.

(4)  Identify the benchmarks and analyze the results.

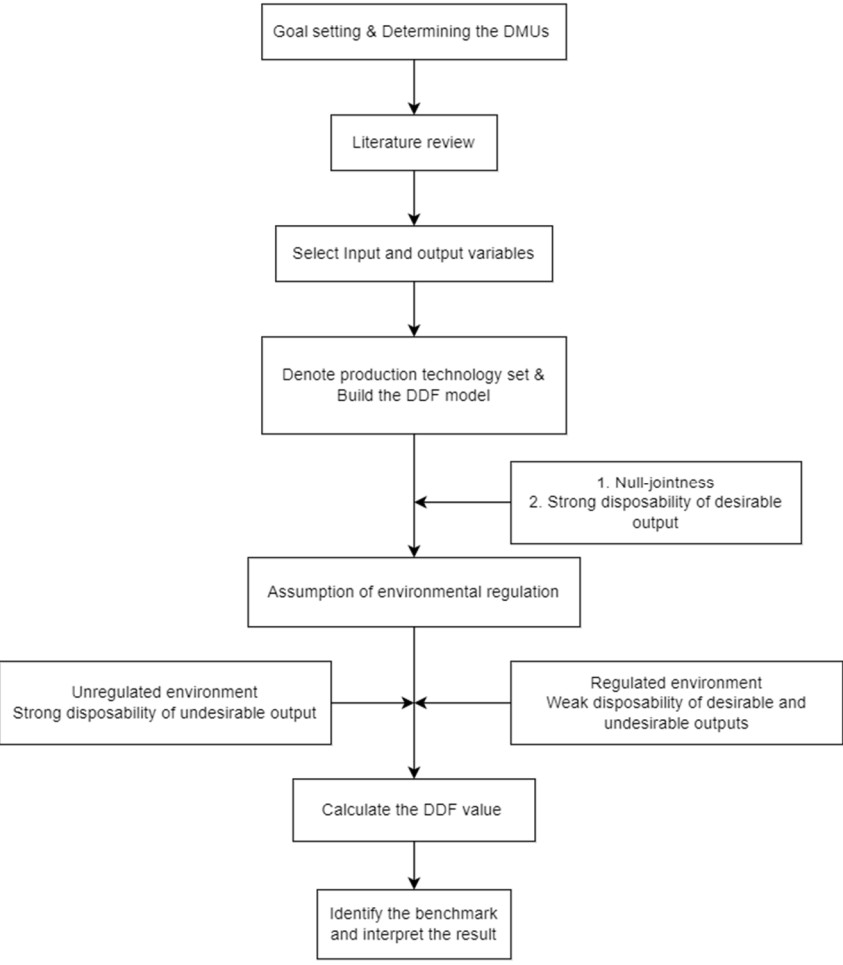

**Figure 1.** Illustration of the proposed DDF process.

## 4. Empirical Results

The DDF value emphasized reducing undesirable output and simultaneously expanding the maximum desired output. Section 3.2 (Model 9) and (Model 10) were used to calculate the DDF values, which represented the distance that the DMUs project into the efficient frontier. When the value = 0, the DMU was located on the production efficiency frontier, and the evaluated unit was efficient. If the value was >0, the evaluated DMU was inefficient. The larger the value, the farther it was from the production efficiency frontier, and the lower the efficiency was. Table 2 presents the results assuming the weak disposability of undesirable output. LSCs reduced undesirable output to comply with environmental regulations. The average DDF values under regulation were 0.2916 in 2019,

0.1869 in 2020, and 0.1238 in 2021. The annual average value was 0.2008, which implies that it was possible to increase desirable output by 20% and reduce undesirable output by 20% without increasing inputs. Of the 11 LSCs, only Wan Hai, HMM, and COSCO achieved DDF values = 0 in 2019. Wan Hai achieved a 0 in 2020 and 2021, as did COSCO in 2021. In the environmental efficiency section, these LSCs represented benchmarks for the other companies. The environmental rating in order was Wan Hai, COSCO, Yang Ming, HMM, ONE, ZIM, Hapag-Llyod, Evergreen, CMA CGM, Maersk, and MSC. In terms of shipping capacity, Wan Hai and COSCO were not large within the DMUs. In contrast, MSC, Maersk, and CMA CGM Group maintained fleets with huge shipping capacities and received low environmental performance ratings. Additionally, the DDF values over three years indicated a decreasing trend. Most LSCs continued to improve their environmental efficiency (see Figure 2).

**Table 2.** DDF value under a regulated environment.

| DMU | Weak Disposability | | | | |
|---|---|---|---|---|---|
| | **2019** | **2020** | **2021** | **Mean** | **Rank** |
| Maersk | 0.4666 | 0.3075 | 0.2405 | 0.3382 | (10) |
| MSC | 0.4781 | 0.3113 | 0.2707 | 0.3534 | (11) |
| CMA CGM Group | 0.3887 | 0.2191 | 0.1440 | 0.2506 | (9) |
| COSCO Group | 0.0008 | 0.0857 | 0 | 0.0288 | (2) |
| ONE | 0.3731 | 0.1543 | 0.0248 | 0.1841 | (5) |
| Hapag-Llyod | 0.3935 | 0.2068 | 0.1329 | 0.2444 | (7) |
| Evergreen Line | 0.3919 | 0.2226 | 0.1233 | 0.2459 | (8) |
| Yang Ming | 0.3415 | 0.0901 | 0.0242 | 0.1519 | (3) |
| Wan Hai Lines | 0 | 0 | 0 | 0 | (1) |
| HMM Co Ltd | 0 | 0.2747 | 0.2485 | 0.1744 | (4) |
| ZIM | 0.3729 | 0.1844 | 0.1533 | 0.2369 | (6) |
| Mean | 0.2916 | 0.1869 | 0.1238 | 0.2008 | |

Table 3 presents the assumption of the strong disposability of undesirable output. In an unregulated environment, environmental regulations did not require a mandatory mitigation of $CO_2$ emissions, and LSCs could release $CO_2$ freely in pursuit of expanding the desired output without increasing investment. Although this assumption does not exist, this paper highlights LSCs using the maximum capacity to increase their desirable output without considering $CO_2$ mitigation. The average values of the DDF were 0.4799 in 2019, 0.5212 in 2020, and 0.3102 in 2021. The annual average value was 0.4371 and LSCs could expand desirable output by 43.71% and contract undesirable output by 43.71%. Wan Hai was efficient in 2019, 2020, and 2021. HMM performed well in 2019 and COSCO also performed well in 2021. The Wilcoxon signed rank test was used to evaluate the directional distance output function of weak and strong disposability, and the significance (*p* value) was 0.005 and less than 0.05, indicating whether the LSCs implement carbon reduction to achieve statistically significant differences before and after $CO_2$ reduction.

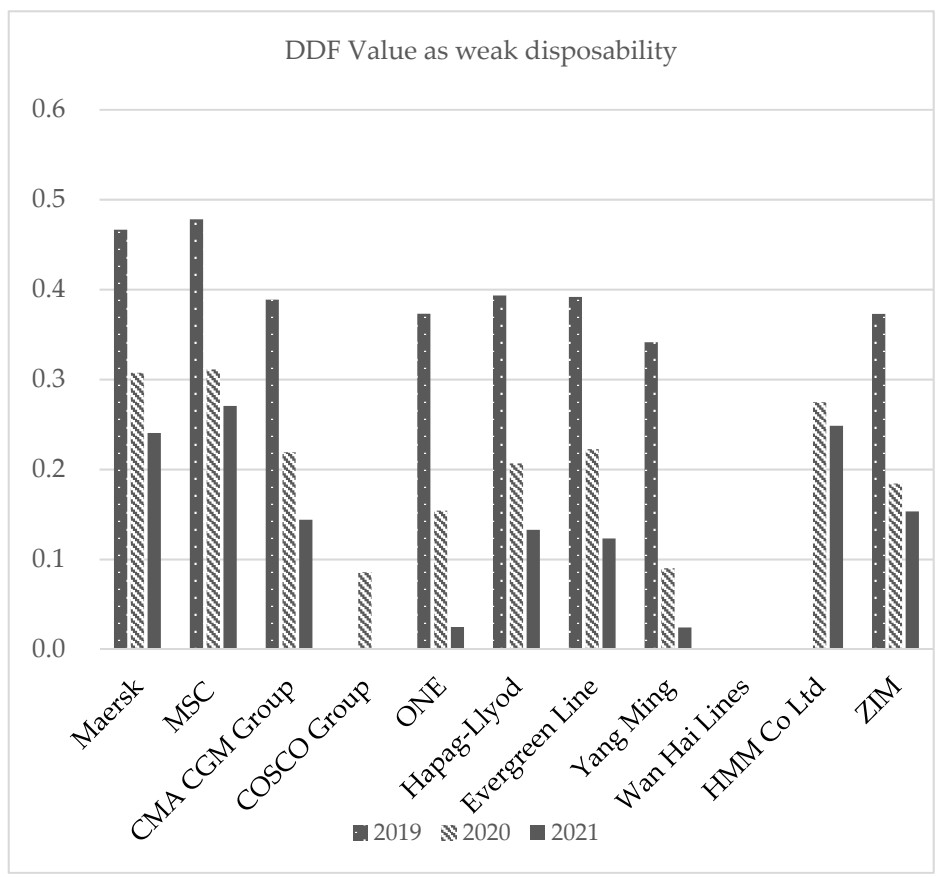

**Figure 2.** Comparison of LSCs' DDF value (2019–2021).

**Table 3.** DDF value in an unregulated environment.

| DMU | Strong Disposability | | | | |
| --- | --- | --- | --- | --- | --- |
| | **2019** | **2020** | **2021** | **Mean** | **Rank** |
| Maersk | 0.7875 | 0.7987 | 0.4814 | 0.6892 | (9) |
| MSC | 0.9160 | 0.9047 | 0.7103 | 0.8437 | (10) |
| CMA CGM Group | 0.5327 | 0.5551 | 0.2976 | 0.4618 | (8) |
| COSCO Group | 0.0008 | 0.1871 | 0 | 0.0626 | (2) |
| ONE | 0.4411 | 0.3192 | 0.0304 | 0.2635 | (4) |
| Hapag-Llyod | 0.5480 | 0.5214 | 0.2685 | 0.4460 | (7) |
| Evergreen Line | 1.2009 | 1.1547 | 0.7717 | 1.0424 | (11) |
| Yang Ming | 0.3738 | 0.1846 | 0.0309 | 0.1964 | (3) |
| Wan Hai Lines | 0 | 0 | 0 | 0 | (1) |
| HMM Co Ltd | 0 | 0.6580 | 0.5090 | 0.3890 | (5) |
| ZIM | 0.4777 | 0.4495 | 0.3120 | 0.4131 | (6) |
| Mean | 0.4799 | 0.5212 | 0.3102 | 0.4371 | |

PAC (pollution abatement cost) implied an opportunity cost. LSCs sacrificed desirable output to reduce undesirable output. RI value was the number of TEUs that LSCs potentially sacrificed the opportunity to carry. The results were led by selected desirable output. Therefore, it was not appropriate to rank LSCs according to RI values. Environmental regulations affect business. Table 4 shows the PAC value, which is the distance between the assumption of weak disposability and the strong disposability of undesirable output. The average PAC were 0.19 in 2019, 0.33 in 2020, and 0.19 in 2021. The annual average PAC value was 0.24. Overall, line shipping potentially reduced 2,654,380 TEUs carried to

comply with environmental regulations. Compared with companies with small capacities, LSCs with large capacities need to reduce more desirable output.

**Table 4.** Regulatory impact value.

| DMU | 2019 DPAC | RI (TEUs) | 2020 DPAC | RI (TEUs) | 2021 DPAC | RI (TEUs) | Mean DPAC | RI (TEUs) |
|---|---|---|---|---|---|---|---|---|
| Maersk | 0.32 | 8,532,882 | 0.49 | 12,412,023 | 0.24 | 6,307,552 | 0.35 | 9,084,152 |
| MSC | 0.44 | 9,196,494 | 0.59 | 13,054,706 | 0.44 | 9,891,010 | 0.49 | 10,714,070 |
| CMA CGM Group | 0.14 | 3,109,632 | 0.34 | 7,056,057 | 0.15 | 3,385,640 | 0.21 | 4,517,109 |
| COSCO Group | 0 | 0 | 0.10 | 1,915,199 | 0 | 0 | 0.03 | 638,400 |
| ONE | 0.07 | 842,538 | 0.16 | 1,969,744 | 0.01 | 67,928 | 0.08 | 960,070 |
| Hapag-Llyod | 0.15 | 1,859,977 | 0.31 | 3,724,456 | 0.14 | 1,614,078 | 0.20 | 2,399,503 |
| Evergreen Line | 0.81 | 5,731,470 | 0.93 | 6,575,516 | 0.65 | 4,809,507 | 0.80 | 5,705,498 |
| Yang Ming | 0.03 | 175,249 | 0.09 | 479,305 | 0.01 | 29,547 | 0.04 | 228,034 |
| Wan Hai Lines | 0 | 0 | 0 | 0 | 0 | 0 | 0 | 0 |
| HMM Co Ltd | 0 | 0 | 0.38 | 1,492,584 | 0.26 | 993,547 | 0.21 | 828,710 |
| ZIM | 0.10 | 294,471 | 0.27 | 753,183 | 0.16 | 609,567 | 0.18 | 552,407 |
| Mean | 0.19 | 2,703,883 | 0.33 | 4,493,889 | 0.19 | 2,518,943 | 0.24 | 2,654,380 |

## 5. Conclusions and Recommendations

The implementation of $CO_2$ emission reduction in the maritime shipping industry is an international concern. As described, existing studies focused primarily on how to mitigate $CO_2$ emission measures, practices, and policies, while the environmental performance of LSCs has rarely been investigated. Therefore, we adopted the DDF to evaluate LSCs' environmental performance. Simultaneously, based on the assumption of the weak disposability and strong disposability of undesirable output, we explored how LSCs potentially sacrificed desirable outputs. First, our findings indicated that LSCs with large shipping capacities ranked low in environmental efficiency. Additionally, the average DDF value was high, far from the production frontier. This implied that most LSCs' overall environmental efficiency was low. These findings are consistent with Gong et al. [30] who evaluated the cargo efficiency of the shipping industry. When the transported cargo was used as desired outputs, most LSCs had low environmental efficiencies compared to other bulk shipping options, and LSCs with huge capacities ranked low in environmental efficiencies. Second, large LSCs had high RI values—for example, MSC and Maersk. It could be said that LSCs transitioning to decarbonization needed to sacrifice desirable outputs. The more capacities these LSCs obtained, the more sacrifices there would be. Therefore, environmental regulations impact these LSCs significantly. Lastly, the annual DDF value was decreasing, moving closer to the production frontier year by year. This indicated that LSCs were moving towards environmental efficiencies. Overall ship $CO_2$ emissions have been lower. The environmental efficiency of LSCs was improving. In short, to achieve carbon reduction targets for LSCs, various carbon reduction measures need to be implemented simultaneously. Due to climate change and increasingly stringent environmental regulations, LSCs must deal with the important issue of reducing $CO_2$ emissions. This paper provided another dimension to examine the undesirable output of the shipping industry, and the findings can provide LSC company management with an understanding of their own company's environmental efficiencies compared to other shipping companies.

This article is limited by the availability of data. Since most shipping firms do not disclose data, only 11 DMUs were selected for this research. Additionally, based on the findings of this paper, Wan Hai has the best environmental efficiency. However, compared with other firms, Wan Hai operates mostly near-ocean routes. Therefore, the configuration of routes may be one of the factors that affect environmental efficiency. Furthermore, as EU legislators have agreed to include maritime transport within the EU emissions trading system (ETS), LSCs face increasing costs. For future research, variables such as route distance, carbon tax, and other undesirable outputs are meaningful regarding the

environmental efficiency of firms. Moreover, only three years of data were collected for this study, so we suggest that the input and output data should continuously be updated in follow-up studies to examine changes in the environmental efficiency of LSCs in the future.

**Author Contributions:** Conceptualization, Y.-H.L.; methodology, H.-S.L.; software, H.-S.L.; validation, Y.-H.L.; formal analysis, Y.-H.L.; investigation, Y.-H.L.; resources, Y.-H.L.; data curation, Y.-H.L.; writing—original draft preparation, Y.-H.L.; writing—review and editing, Y.-H.L. and H.-S.L.; visualization, Y.-H.L.; supervision, H.-S.L. All authors have read and agreed to the published version of the manuscript.

**Funding:** This research received no external funding.

**Institutional Review Board Statement:** Not applicable.

**Informed Consent Statement:** Not applicable.

**Data Availability Statement:** Not applicable.

**Conflicts of Interest:** The authors declare no conflict of interest.

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
