# Peer review of "Using a Directional Distance Function to Measure the Environmental Efficiency of International Liner Shipping Companies and Assess Regulatory Impact"

_sustainability, doi:10.3390/su15043821_

Round 1
Reviewer 1 Report
The article addresses the highly topical issue of decarbonization of maritime transport. The content and the research approach are important for the development of a scientific approach to decarbonization of transport. The article has some shortcomings that the authors need to address.
Introduction
The introduction is very comprehensive, but does not address all approaches to reducing the carbon footprint of maritime transport. For example, slow steaming/speed, which emits less CO2 for the same transport distance, is not discussed. Ship owners also use up to 40% slow steaming. It is also proven that smaller and older ships emit more CO2 per TEU transported. In addition, indirect services cause higher emissions due to the additional handling of containers in ports.
Literature review
The chapter is very long, but it is a kind of interweaving of previous studies on the topic of CO2 emissions in maritime liner services. In lines 110-111, the authors summarise 6 sources in a very general sentence with efficiency assessments, without comparing common findings and differences. In lines 116-134, they discuss only one source, which is incorrectly cited in line 124 (reference in parentheses). Also, the source citations in lines 138-143 and in line 183 do not comply with the journal's instructions (years of sources - square brackets missing).
I suggest adding relevant research to the source references to show gaps in the methodological approach to CO2 reduction in the maritime industry, e.g., https://www.mdpi.com/2077-1312/10/11/1644 - - https://www.mdpi.com/2077-1312/10/7/873 https://doi.org/10.1016/j.ocecoaman.2021.105936
Methodology
The methodology is mainly based on the definition of the DDF approach and lacks the overall methodological approach to conduct the research. I suggest a graphical illustration of the methodological steps that would make it easier for other researchers to reproduce the methodological approach. It would be good to briefly define each step from a resourcing perspective and briefly describe the research approaches used.
Conclusion
There is a lack of justification as to why there are differences between the shipping companies. MSC and Wan Hai are specifically mentioned. How do the other shipowners perform? Why are there discrepancies between them? It would be necessary to find and argue different research results of shipping companies. It would be appropriate to compare the results obtained with other studies/results already conducted. Moreover, the sentence in lines 394-395 is already common knowledge, as well as the one in lines 401-402.
In some places, the text needs to be grammatically corrected.
Reviewer 2 Report
The manuscript deals with the issue of assessing the environmental efficiency of international liner shipping companies using the directional distance function. The topic of the study is very important and timely. The manuscript is well-written, precise, and easy to understand. Based on the results of previous studies, the authors well justified the choice of research method. The research method is exhaustively described. In the study, the authors analyze the main liner shipping companies, which account for more than 80% of the world's shipping capacity. The research sample is representative. The authors present valuable results of the research. The obtained results indicate that although the overall environmental efficiency of liner shipping is low, liner shipping companies are moving towards carbon reduction goals, and overall environmental efficiency is improving year by year. The results of the study also showed that environmental regulation had a significant impact on liner shipping companies. The significance of the findings in light of what was already known about the research problem being investigated is well underlined in the conclusions. Managerial implications are clearly listed and described. The authors also highlight the limitations of the research results (data availability, reference period) and directions for further research.
Author Response
We would like to thank you for providing your constructive and detailed review comments on our manuscript. The recommendations and advice have helped us to significantly enhance the quality of the manuscript.
Reviewer 3 Report
Abstract
- Page 1/Lines 13-14: CO2 causes 'climate change' not 'clime change'.
- Page 1/Line 14-16: IMO aims to reduce GHGs by 70% in 2050 compared to 2008, not 50%.
Introduction
- Page 1/Line 32: The first sentence should be rewritten. As it stands, it is difficult to understand.
- Page 2/Line 76: Please provide full forms of gases at first presence.
- At the end of the Introduction, the authors indicated that 'there has been little research on liner shipping companies' environmental efficiency performance'. However, a quick search (with only one keyword 'liner shipping emission') on WoS shows that there are 151 studies focusing on the issue. In Literature Review, the authors provide a piece of knowledge of various methods for measurement of environmental efficiency but the readers cannot understand why this study is needed and what the literature gap is. Please provide a more detailed literature review and state the novelty of this paper in the Introduction.
Data Sources and Empirical Findings
- Section 4.2 must be presented as a different section as Results and Section 4.1 may be presented as a part of the Methodology. The Methodology title may be changed to Materials and Methodology.
- It is not clear how the authors have reached the results and discussions on the results are missing. The authors must interpret the results with the results of previous studies.
Conclusions and Recommendations
- Page 10/Lines 385-386: The authors indicate that 'this study differs from previous literature ...', however, they don't present any comparable knowledge to the readers. Please see the above-mentioned comments.
Round 2
Reviewer 3 Report
The authors revised the manuscript in accordance with the comments.